# Dysregulation of erythropoiesis and altered erythroblastic NMDA receptor-mediated calcium influx in Lrfn2-deficient mice

**Ryuta Maekawa**[1], **Hideki Muto**[2], **Minoru Hatayama**[1], **Jun Aruga**[1]*

**1** Department of Medical Pharmacology, Nagasaki University Institute of Biomedical Sciences, Nagasaki, Japan, **2** Biomedical Research Support Center, Nagasaki University School of Medicine, Nagasaki, Japan

* aruga@nagasaki-u.ac.jp

## Abstract

*LRFN2* encodes a synaptic adhesion-like molecule that physically interacts with N-methyl-D-aspartate (NMDA) receptor 1 and its scaffold proteins. Previous studies in humans and mice have demonstrated its genetic association with neurodevelopmental disorders such as learning deficiency and autism. In this study, we showed that Lrfn2-deficient (KO) mice exhibit abnormalities of erythropoietic systems due to altered NMDA receptor function. In mature Lrfn2 KO male mice, peripheral blood tests showed multilineage abnormalities, including normocytic erythrocythemia, and reduced platelet volume. Colony forming unit assay using bone marrow cells revealed decreases in the counts of erythrocyte progenitors (CFU-E) as well as granulocytes and monocyte progenitors (CFU-GM). Whole bone marrow cell staining showed that serum erythropoietin (EPO) level was decreased and EPO receptor-like immunoreactivity was increased. Flow cytometry analysis of bone marrow cells revealed increased early erythroblast count and increased transferrin receptor expression in late erythroblasts. Further, we found that late erythroblasts in Lrfn2 KO exhibited defective NMDA receptor-mediated calcium influx, which was inhibited by the NMDA receptor antagonist MK801. These results indicate that Lrfn2 has biphasic roles in hematopoiesis and is associated with the functional integrity of NMDA receptors in hematopoietic cells. Furthermore, taken together with previous studies that showed the involvement of NMDA receptors in hematopoiesis, the results of this study indicate that Lrfn2 may regulate erythropoiesis through its regulatory activity on NMDA receptors.

## Introduction

Lrfn2 (also known as SALM [synaptic adhesion like molecule] 1) is a member of the Lrfn family proteins, which are known to be cell adhesion molecules that regulate neuronal and synaptic development in the brain [1]. Lrfn2 physically interacts with ion channels forming a glutamate receptor called N-methyl-D-aspartate receptor 1 (NMDAR1, also known as GluN1 or Grin1) [1] as well as a synaptic scaffold protein PSD95 (post synaptic density protein 95, also known as Dlg4) [1–3]. Recent genetic studies have linked human *LRFN2* to autism-

**Funding:** MEXT KAKENHI 16H04666, 19H03327, 20K21605 Smoking Research Foundation The Uehara Memorial Foundation.

**Competing interests:** The authors have declared that no competing interests exist.

spectrum disorders [4], working memory deficits [5], and anti-social personality disorder [6]; moreover, mouse Lrfn2 deficiency results in altered synaptic plasticity and abnormal social behaviors [4,7].

The above studies addressed the roles of the Lrfn2 family proteins in the nervous system, whereas Castellanos et al. [8] investigated the role of the Lrfn2 protein in hematopoiesis. According to their study, Lrfn2 overexpression, in a colony formation assay using murine bone marrow (BM) cells, increased erythroid progenitor-containing colonies (BFU-E) as well as colonies containing common progenitors for granulocyte/erythrocyte/macrophage/mega-karyocyte (CFU-GEMM), suggesting increased erythropoiesis due to Lrfn2 overexpression [8]. They also reported that Lrfn2 overexpression in BM cells causes the overgrowth of an unclassifiable "bizarre" cell type [8]. These results, collectively, revealed that overexpressed Lrfn2 could affect hematopoiesis. However, the physiological significance of Lrfn2 remains unclear due to the absence of loss-of-function analysis. Moreover, there have been no mechanistic models to explain the Lrfn2-mediated control of hematopoiesis.

On the other hand, recent studies revealed that NMDA receptors play significant roles in hematopoiesis [9]. NMDA receptor subunits are expressed in megakaryocytes and erythroid cells and function as ion channels, similar to their neuronal counterparts [9]. NMDAR1 deficiency impairs differentiation of megakaryocytic-erythroid progenitor (MEP)-like leukemia cells (Meg-01 cells) [10]. It is proposed that the NMDA receptors balance both megakaryocytic and erythroid cell fates at the level of a bipotential MEP [9]. These facts attract our attention to the functional relationship between Lrfn2 and NMDA receptors in hematopoiesis.

In this study, we examined the hematopoietic phenotype of Lrfn2-deficient (knockout, KO) mice. Our results indicated that Lrfn2 is involved in the control of erythropoiesis. We also found dysregulation of NMDA receptor function in late erythroblasts and discussed a potential mechanism to explain the Lrfn2-mediated erythropoiesis control.

## Materials and methods

### Animals

All animal experiments were approved by the Animal Care and Use Committee of Nagasaki University (approval number 1803271441–5) and carried out in accordance with the guidelines for animal experimentation at Nagasaki University. Lrfn2 KO (*Lrfn2*$^{-/-}$) mice were created and genotyped as described [4], and were backcrossed to C57BL/6J mice for more than ten generations before commencement of the experiments. Adult male mice 3–27 M-old were used in this study. Mice were maintained on a 12-h dark/light cycle (7 a.m./7 p.m.) with food and water provided ad libitum. Mice older than 7 M were subjected to behavioral assays before the study. The behavioral tests included open field test, social discrimination test, and fear-conditioning test, all of which were carried out equally for age-matched Wild type (WT) and KO mice. WT littermates (*Lrfn2*$^{+/+}$) are used as the control group in all analyses. A total of 67 WT and 67 KO mice were used in this study. Each sample was derived from a single mouse without pooling. Common mice were used partly between the peripheral blood test and the other tests. Common mice were used between spleen weight measurement and (calcium measurement test or colony assay). Sample sizes for each experiment were determined such that the power and significance in two sided-test were 80% and 5%, respectively [11]. However, the number of samples from animals were minimized empirically.

### Peripheral blood test

Peripheral blood (100–200 μL) was collected from a mouse at 3M (2.0–4.3 M-old), 6M (5.8–7.2 M-old), and 12M (10.3–13.1 M-old) after birth, by means of cheek bleeding [12], and

stored in BD microtainer tubes (Beckton, Dickinson and Company, Franklin Lakes, NJ) containing EDTA. Some mice were repeatedly bled at multiple stages. Hematological profiles were determined using a veterinary hematology analyzer (thinka CB-1010, ARKRAY, Kyoto, Japan).

### Measurement of serum EPO

Sera (20–60 μL) were prepared from the peripheral blood collected from 12–14 M-old mice by cheek bleeding and stored at -80˚C. The samples were diluted six-fold, and EPO measurement was done using Quantikine ELISA Mouse Erythropoietin (R&D Systems, Minneapolis, MN), according to manufacturer's instructions. All samples were measured in duplicate.

### Colony forming unit (CFU) assay

Femurs were isolated from 3–6 M-old mice (BM cells) or 11–12 M-old mice (spleen cells). Cells were plated in methyl cellulose MethoCult medium (M3434, M3234, STEMCELL Technology, Vancouver, Canada), as per the manufacturer's instructions. For BFU-E, CFU-GM, and CFU-GEMM, $1.0 \times 10^4$ BM cells or $4.0 \times 10^5$ spleen cells obtained from a mouse were cultured in two 35 mm dishes in the presence of SCF, IL-3, IL-6, and EPO (M3434) for 12 or 13 days before observation. For CFU-E, $6.8 \times 10^5$ BM or $4.0 \times 10^5$ spleen cells obtained from a mouse were cultured in three 35 mm dishes containing MethoCult medium (M3234) supplemented with 10 U/mL rhEPO for 2 days before observation. BM cells were not treated to lyse red blood cells. Images of the entire 35-mm dish field were obtained using the image-stitching function of the BZ-X800 microscope (Keyence, Osaka, Japan). Colonies were counted by an observer who was blinded to the genotypes.

### Flow cytometric analysis

For the analysis of erythroid progenitors in BM, CD71/TER119 double labeling was performed as described [13,14]. BM cells were isolated from femurs of 18–22 M-old mice and washed with 5% fetal bovine serum-containing Dulbecco's phosphate buffered saline without calcium and magnesium (5% FBS-PBS). After blocking the Fc receptor using anti-mouse CD16/32 antibodies (Biolegend, San Diego, CA), BM cells were incubated on ice for 1 h with FITC-Ter119 (Biolegend) and PE-CD71 (Biolegend) after blocking Fc receptors. Control samples were incubated with the FITC-IgG2b and PE-IgG1 isotype control antibodies (Biolegend). 7AAD dye (COSMO BIO, Tokyo, Japan) was added 5 min before flow cytometry analysis to exclude dead cells. Flow cytometry was performed using the BD LSRFortessa X-20 cell analyzer (BD Biosciences). All Ter119-positive cells were classified into four subsets: ProE (Ter119$^{med}$CD71$^{high}$), EryA (Ter119$^{high}$CD71$^{high}$FSC$^{high}$), EryB (Ter119$^{high}$CD71$^{high}$FSC$^{low}$), and EryC (Ter119$^{high}$CD71$^{low}$FSC$^{low}$), corresponding to morphologically recognized proerythroblasts and basophilic, polychromatic, and orthochromatic erythroblasts, respectively.

### Intracellular calcium concentration measurement

Calcium concentration of BM cells was measured using flow cytometry, according to a method based on the one described by Hanggi et al. [15]. BM cells were isolated from femurs of 6–12 M-old or 19–22 M-old mice and washed with 5% FBS-PBS. Live cell numbers were counted using the TC20 cell counter (Bio-Rad, Hercules, CA) with trypan blue staining. BM cells ($1 \times 10^6$) were suspended in 0.1 mL of erythroid progenitor culture medium [16] consisting of Iscove's modified Dulbecco's medium (IMDM) containing 0.51 mM glutamic acid, 0.40 mM glycine, 1.5 mM calcium chloride (FUJIFILM Wako, Osaka, Japan), 15% FBS (Thermo Fisher

Scientific, Waltham, MA), Insulin-Transferrin-Selenium (ITS-G, Thermo Fisher Scientific), 0.45 mM $\alpha$-monothioglycerol (FUJIFILM Wako), and 50 µg/ml ascorbic acid (Sigma-Aldrich, St. Louis, MO). The cells were then loaded with 3 µM Fluo4-AM (DOJINDO LAB, Kumamoto, Japan) for 30 min at 37˚C, followed by further 30 min incubation at 37˚C with the following antibodies: APC-Ter119 (Biolegend) and PE-CD71 (Biolegend). For NMDA receptor antagonist treatment, MK-801 (final 80 µM, Cayman Chemical, Ann Arbor, MI) was added to the cell suspensions just before antibody addition and incubated for 30 min at 37˚C. The cell culture medium was replaced by a flow cytometry solution (pH 7.35) containing (in mM) 135 NaCl, 5 KCl, 5 HEPES, 10 D-glucose, and 2 $CaCl_2$. The cells were washed twice with the flow cytometry solution before measurement. Agonist-induced $Ca^{2+}$ uptake was recorded as the response to the administration of 150 µM NMDA and 50 µM glycine (NMDA/GLY) to the flow cytometry solution containing the cells. The time window for the measurement was 7 to 37 sec after the NMDA/GLY addition. All assays were performed in duplicate.

## Immunostaining analysis

BM cells were fixed with 4% paraformaldehyde for 20 min at room temperature and the cells were rinsed and suspended in the culture medium. $1 \times 10^6$ cells were pelleted in microfuge and resuspended in 0.2 mL of deionized $H_2O$. 0.1 mL of the cell suspension was added to a chamber (48 $mm^2$) in 3-chamber cytofuge cassette (StatSpin Cytofuge 2, Nihon Rufuto, Tokyo, Japan) on MAS-coated slide glass (Matsunami Glass, Osaka, Japan), and settled onto glass surface by centrifugation at 1000 rpm for 4 min. The cells were briefly dried on a hot plate at 30 ˚C. The cells were then immersed in PBS, incubated in a blocking buffer (5% donkey serum, 0.1%TX100, PBS) for 45 min at room temperature, and reacted with goat anti mouse EpoR antibody (AF1390, R&D Systems, 1/200 in the blocking buffer) at 4˚C overnight. The primary antibody was detected by Alexa488-conjugated donkey anti-goat IgG (Jackson ImmunoResearch, West Grove, PA). After mounted in VectaShield with DAPI (Vector Laboratories), images were taken by LSM800 confocal microscope (ZEISS, Oberkochen, Germany). Images were analysed by using Image J (https://imagej.nih.gov/ij/index.html).

## Statistical analysis

All data were expressed as the mean ± standard deviation (SD). Statistical analyses of all data were performed using two tailed Student's *t*-test for two group comparisons, unless otherwise noted. Two-way analysis of variance was used to evaluate the effect of MK-801. Percentage values in the results section indicate the percentage of (KO mean value -WT mean value) when WT mean values are defined as 100%. A *P* value of less than 0.05 was considered statistically significant.

## Results

### Hematological abnormalities in Lrfn2 KO mice

To address the role of Lrfn2 in hematopoiesis, we first determined the hematological profiles of Lrfn2 KO mice in the 3 M, 6 M, and 12 M old groups in comparison to littermate WT mice. The results (Fig 1, S1 Table) revealed multilineage abnormalities. With regards to erythrocyte-related indexes, erythrocyte numbers (Fig 1A), hemoglobin (Fig 1B), and hematocrit (Fig 1C) were increased in both 6 M (11.3%, 11.2%, and 12.0%, respectively) and 12 M (6.4%, 7.7%, and 6.7%, respectively) old groups; moreover, 3 M old mice showed the same tendency (6.7%, 5.3%, and 6.2%, respectively). No significant difference in the mean corpuscular volume was observed among the groups, indicating normocytic erythrocythemia in Lrfn2 KO mice. However, mean corpuscular hemoglobin level in 3 M old Lrfn2 KO mice was slightly (1.5%)

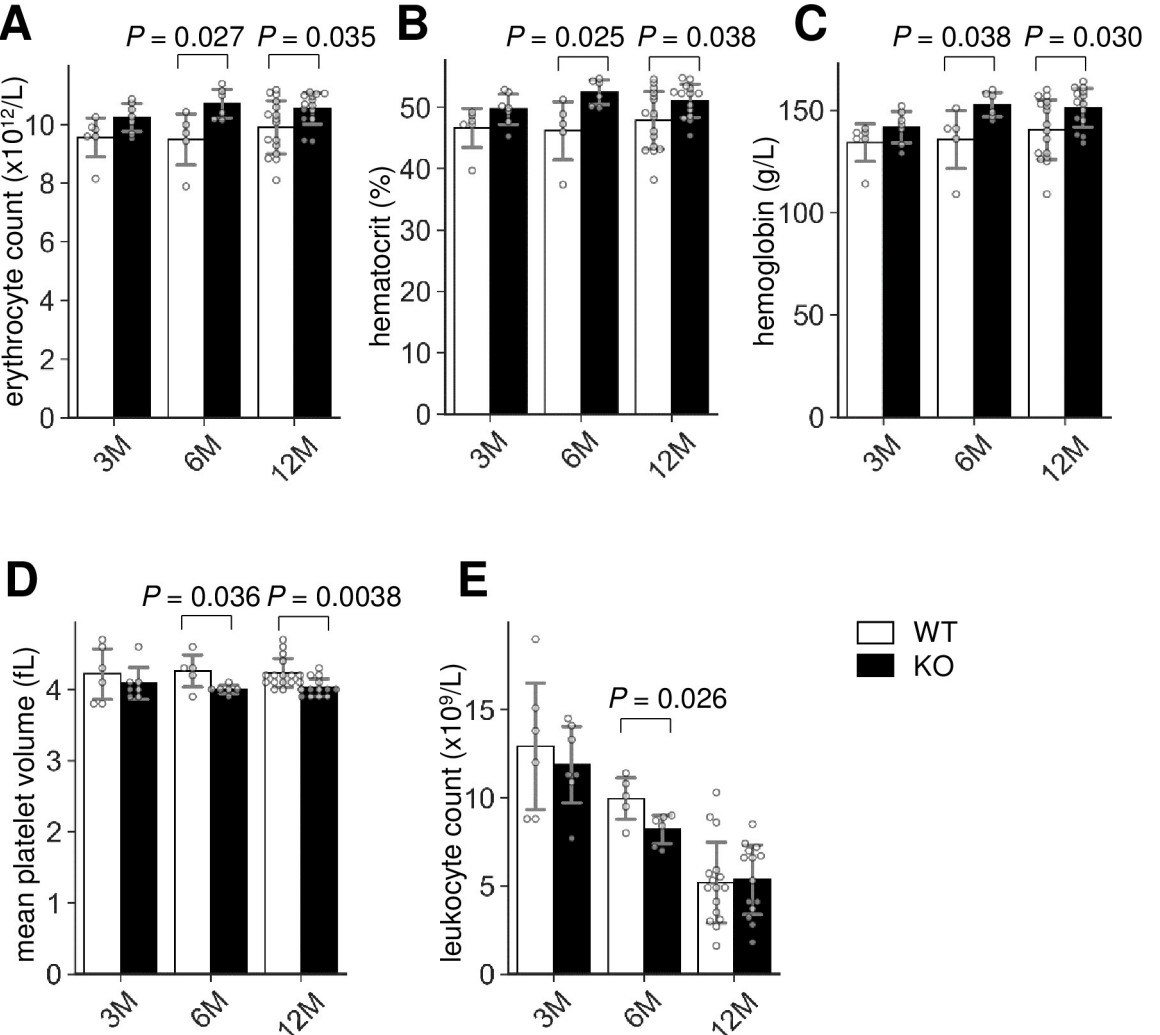

**Fig 1. Hemogram of Lrfn2 KO mice.** Complete blood count of male Lrfn2 WT (open bars) and KO (closed bars) mice aged 3 M (2.0–4.3 M-old; WT, $n$ = 6; KO, $n$ = 7), 6 M (5.8–7.2 M-old; WT, $n$ = 5; KO, $n$ = 6), and 12 M (10.3–13.1 M-old; WT, $n$ = 16; KO, $n$ = 14). (A) Erythrocyte count. (B) Hematocrit. (C) Hemoglobin. (D) Mean platelet volume. (E) Leukocyte count. Additional results are indicated in S1 Table. *Open bar*, WT; *closed bar*, KO; *error bar*, SD. Each value from a mouse is indicated by *circles*. P values indicate those obtained by two tailed *t*-tests.

reduced in comparison to WT mice. In the context of platelet-related indexes, reduced platelet volumes were observed in both 6 M (- 6.5%) and 12 M (- 4.7%) old mice (Fig 1D). Among leukocyte-related indexes, the total leukocyte number was decreased (- 21%) in 6 M old mice (Fig 1E), with reduced counts of granulocytes and lymphocytes (S1 Table).

## Colony forming units were reduced in Lrfn2 KO

To investigate whether any relevant phenotypes exist in the BM of Lrfn2 KO mice, we first performed colony forming unit assay. BM cells were cultured in methyl cellulose medium and the resultant colonies were counted. As shown in Fig 2, the number of granulocyte/macrophage-colony forming units was decreased in the Lrfn2 KO group (- 26%, $P$ = 0.012). The erythroid-burst forming unit (BFU-E) showed a marginal decrease (- 17%, $P$ = 0.27), and the erythroid-colony forming unit (CFU-E) was significantly decreased in comparison to WT group (- 20%,

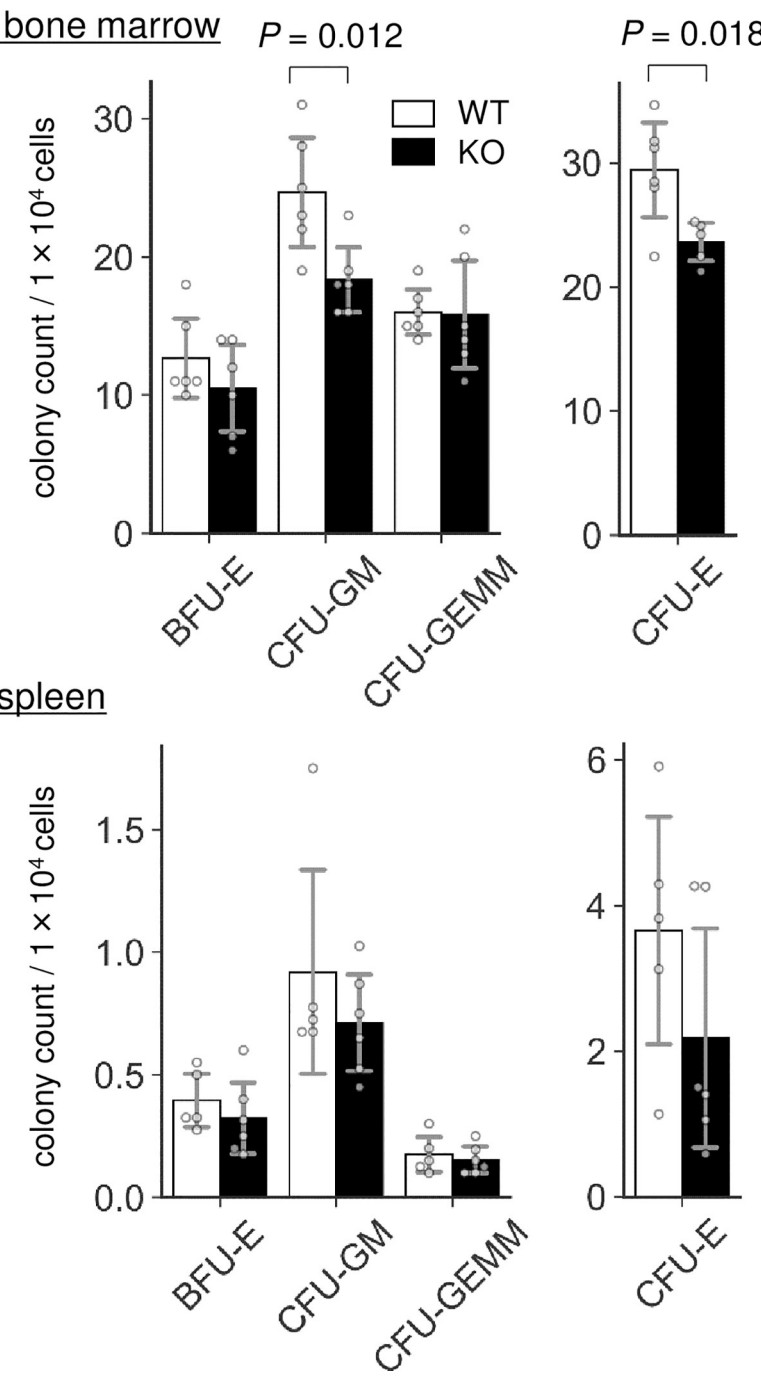

**Fig 2. Colony forming unit (CFU) assay.** BM cells from 3–6 M-old male WT ($n = 6$) and Lrfn2 KO ($n = 5$ or 6) mice and spleen cells from 11–12 M-old male WT ($n = 5$) and Lrfn2 KO ($n = 6$) mice were used for the assay. Mean colony numbers in a dish are indicated. *Open bar*, WT; *closed bar*, KO; *error bar*, SD. Each value from a mouse is indicated by *circles*. *P* values indicate those obtained by two tailed *t*-tests.

$P = 0.018$). We also performed colony forming unit assay using spleen cells and found that CFU-E tended to decrease (- 40%, $P = 0.18$) in Lrfn2 KO mice.

Both BFU-E cells and CFU-E cells represent two differentiation stages in early erythropoie-sis [pro-erythroblasts, early (basophilic) erythroblasts, mid (polychromatic) erythroblasts, and

late (orthochromatic) erythroblasts]. BFU-E cells require both stem cell factors and EPO whereas CFU-E cells require only EPO [17]. A recent study involving transcriptome analysis showed that BFU-E and CFU-E represent CD45⁺GPA⁻IL-3R⁻CD34⁺CD36⁻CD71^low cells and CD45⁺GPA⁻IL-3R⁻CD34⁻CD36⁺CD71^high cells, respectively, and suggested that erythroid differentiation progresses in the following order: CD34⁺ hematopoiesis progenitor cells → BFU-E cells → CFU-E cells → pro-erythroblasts [17]. Thus, the colony forming unit assay demonstrated that Lrfn2 KO mice exhibited either reduced counts of BFU-E cells or accelerated differentiation to pro-erythroblast, or both, irrespective of the increased count of mature erythrocytes. In any case, it was thought that early phase of erythropoiesis was impaired in Lrfn2 KO mice.

## Serum EPO level was reduced in Lrfn2 KO mice

As erythrocythemia can be secondarily caused by increased EPO levels [18], we measured the serum EPO levels in Lrfn2 KO mice (Fig 3A) and found that it was significantly decreased (-6.7%, $P = 0.022$). Therefore, erythrocytosis in Lrfn2 KO mice was considered to be caused by the alteration of a mechanism intrinsic to erythroid progenitors. Interestingly, 10–12 M-old Lrfn2 KO mice showed significant body weight loss (-14%, $P = 0.00054$) (Fig 3B), which was severer than that observed in 8 weeks-old mice (- 6.0%, $P = 0.022$) [4]. Because splenomegaly is included in the diagnostic criteria of polycythemia vera [19], we measured the weight of the spleen (Fig 3B). In both the 10–12 M-old and 19–27 M-old age groups, the average weights of spleens were comparable between WT and KO mice (Fig 3B). However, the weights of spleen from KO mice were less divergent than from WT mice in the 10–12 M-old age group ($P = 9.0 \times 10^{-5}$ in f-test) but more divergent in the 19–27 M-old age group ($P = 0.041$ in f-test) (Fig 3B), indicating hematological dysregulation that is distinct from polycythemia vera. The increased serum EPO level, together with Lrfn2 overexpression [8], led us to hypothesize that Lrfn2 deficiency directly affects erythropoiesis *in vivo*.

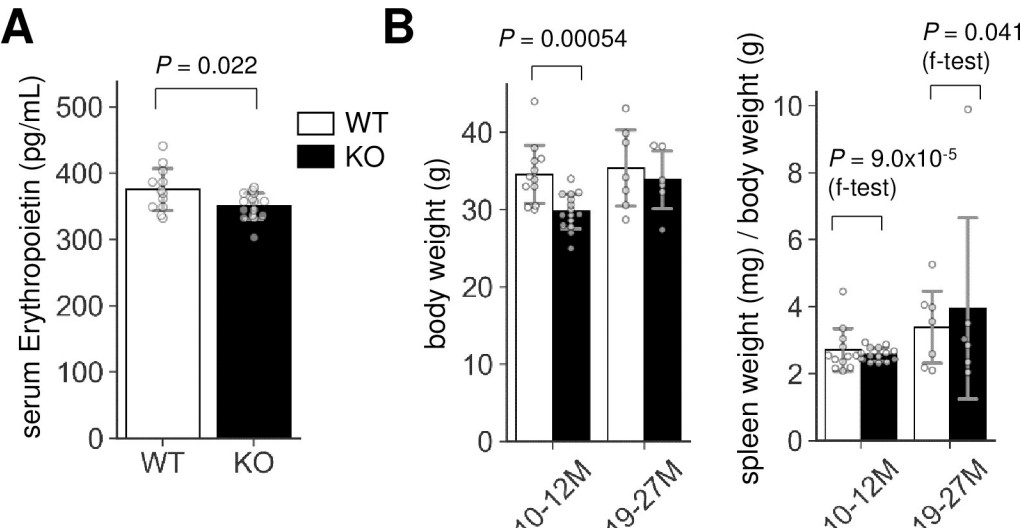

**Fig 3. Erythropoietin and spleen weight abnormalities in Lrfn2 KO.** (A) Erythropoietin levels in 12–14 M-old Lrfn2 WT ($n = 12$) and KO ($n = 15$) mice. (B) Body weight (left) and spleen weight normalized to the body weight (right) of Lrfn2 KO and WT mice aged 10–12 M-old (left; WT, $n = 12$; KO, $n = 15$) or 19–27 M-old (right; WT, $n = 7$; KO, $n = 6$) stage. *Open bar*, WT; *closed bar*, KO; *error bar*, SD. Each value from a mouse is indicated by *circles*. P values indicate those obtained by two tailed *t*-tests (A) or f-tests (*f-test* in B).

## Abnormalities of the late erythropoiesis in Lrfn2 KO mice

We next examined the erythrocyte progenitors in the BM of Lrfn2 KO mice using flow cytometry. Double labeling with CD71, a transferrin receptor, and TER119, a cell surface antigen specific to mature erythroid cells, enabled us to define subsets of erythroid cell lineage [13]. We investigated Lrfn2 KO and WT BM cells from 6–12 M and 18–22 M old mice (Fig 4, S2 Table). In terms of cell counts, the number of early erythroblasts (EryA) in the Lrfn2 KO group was larger (12%, $P = 0.0089$) than that in the WT group, whereas the number of total erythroid lineage cells (TER119+ cells) was comparable in both groups (Fig 4B, S2 Table). Furthermore, the expression of CD71 in late erythroblasts (EryC) was higher in the Lrfn2 KO group than in the WT group (21%, $P = 0.0037$) (Fig 4D, S2 Table). Proerythroblast (proE) side scattering (- 8.3%, $P = 0.060$) tended to be lower in Lrfn2 KO (Fig 4C, S2 Table). These results suggested that the late phase of erythropoiesis was affected by the lack of Lrfn2.

## Altered NMDA receptor-mediated calcium influx in Lrfn2 KO erythroid progenitors

We further investigated the mechanism by which Lrfn2 controls erythropoiesis. We hypothesized that NMDA receptor was involved in Lrfn2-regulated erythropoiesis because previous studies have demonstrated the NMDAR1-binding ability of Lrfn2 [1] and the essential roles of NMDA receptors in human erythroid precursor cells [15]. To examine the function of NMDA receptor in Lrfn2 KO BM cells, we measured the intracellular $Ca^{2+}$ level using the membrane permeable $Ca^{2+}$-sensitive dye Fluo-4 AM. BM cells were loaded with Fluo-4 AM, labeled with TER119 and CD71 antibodies, and stimulated with the NMDA receptor agonists NMDA and glycine (NMDA/GLY) immediately before flow cytometry (Fig 5A).

NMDA receptor-induced changes in $Ca^{2+}$ level and the reduction of $Ca^{2+}$ level caused by pretreatment with the NMDA receptor non-competitive antagonist MK-801 were measured for two age groups (6–12 M and 19–22 M) in mice (Fig 5B). In WT BM cells, NMDA-induced changes were observed in the $Ca^{2+}$ levels of mid erythroblasts (EryB, 19–22 M) and late erythroblasts (EryC, 6–12 M and 19–22 M). MK-801-induced suppression was observed in late erythroblasts (EryC, 6–12 M), in which NMDA/GLY induced lower ΔF/F in KO BM cells (mean ± SD, 106 ± 119%) than in WT cells (mean ± SD, 304 ± 158%) (6–12 M, $P = 0.013$); notably, the effects of MK-801 on the NMDA/GLY response were different between KO cells and WT cells (19–22 M, $P = 0.019$).

Moreover, the effects of MK-801 were different between WT and KO cells in the 6–12 M group ($P = 0.012$ in two-way ANOVA of ΔF/F, including genotype [WT, KO] and treatment [NMDA/GLY, NMDA/GLY+MK801]). In mid erythroblasts (EryB) of the 19–22 M age group, the variance of NMDA/GLY was larger in KO cells than that in WT cells ($P = 0.0027$ in f-test). These results indicated that Lrfn2 KO BM cells had impaired NMDAR function in late erythroblasts, consistent with the larger NMDA/GLY response variance (6–12 M, $P = 0.0063$ in f-test) and MK-801-induced suppression variance (19–22 M, $P = 0.021$ in f-test) in WT erythroid cells (TER119+) compared to those in KO erythroid cells; moreover, the NMDA receptor functions were limited to late (6–12 M and 19–22 M) or mid (19–22 M) erythroblasts.

In addition, $Ca^{2+}$ analysis uncovered an aging effect on the NMDA receptor functions in mouse erythroid progenitors (Fig 5B). When compared between 6–12 M and 19–22 M WT erythroblasts, NMDA/GLY-induced ΔF/F was found to be lower in the 19–22 M group (mean ± SD, 26 ± 31%) than in the 6–12 M group (mean ± SD, 304 ± 158%, $P = 0.0015$) whereas the corresponding difference in the KO group was not so clear ($P = 0.057$).

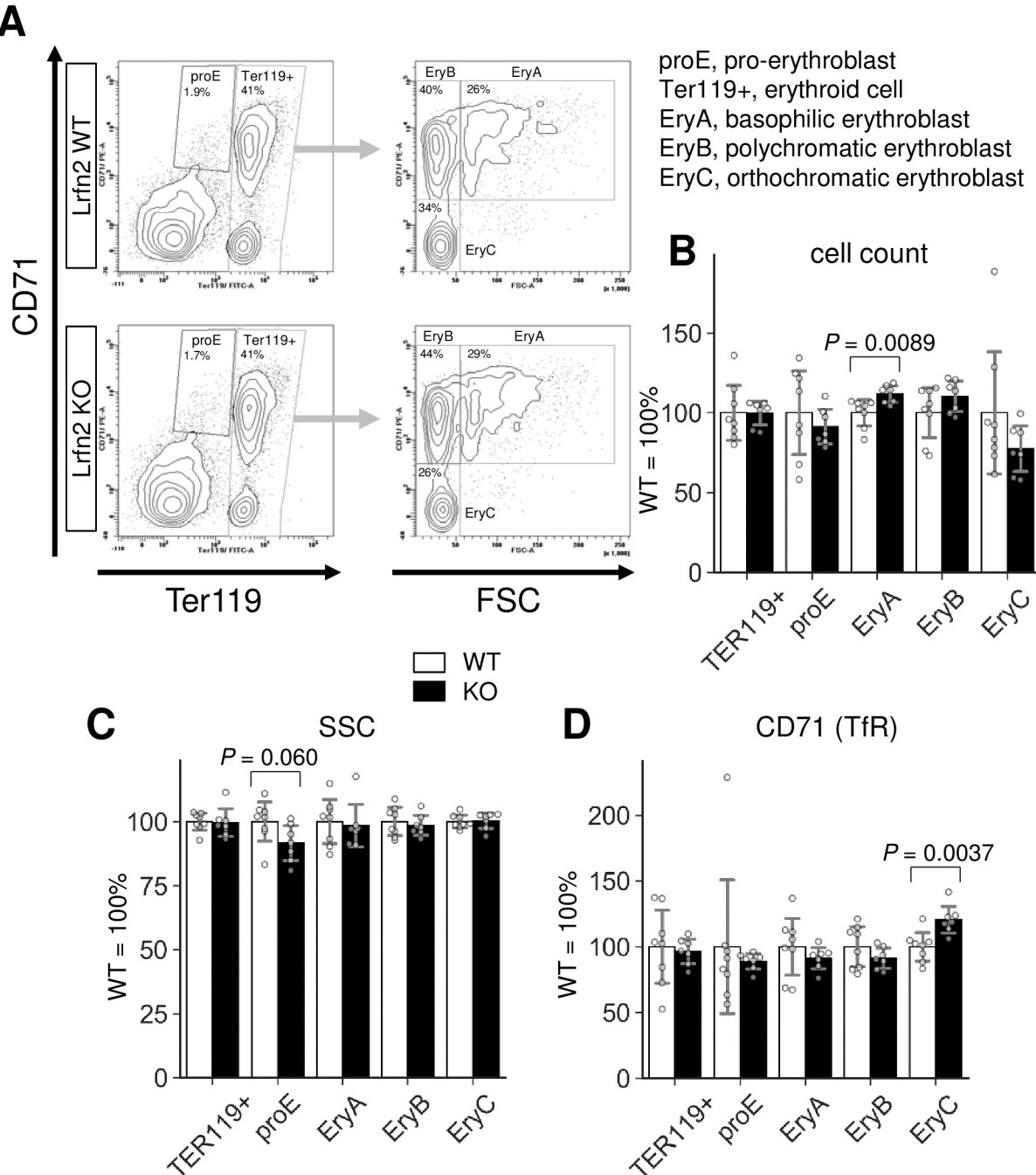

**Fig 4. Flow cytometry analysis of Lrfn2 KO BM cells.** BM cells from 18–22 M-old Lrfn2 WT ($n = 8$) and KO ($n = 7$) mice were subjected to CD71/TER119 flow cytometry. (A) Gating strategy. (B) Cell count, number of cells out of 10,000 living cells (7AAD-negative). (C) SSC, side scatter area. (D) CD71, mean immunofluorescence intensity (arbitrary unit) of anti-CD71-PE. TER119, mean immunofluorescence intensity (arbitrary unit) of anti-Ter119-FITC; FSC, forward scatter area. In B-D, *Open bar*, WT; *closed bar*, KO; *error bar*, SD. Each value from a mouse is indicated by *circles*. *P* values were obtained by two tailed *t*-tests Additional results are indicated in S2 Table.

## Erythropoietin receptor protein level was increased in Lrfn2 KO BM cells

To further clarify the basis of erythrocythemia in Lrfn2 KO cells, we examined EPO receptor (EpoR) expression. Immunostaining of BM cells revealed significant increase in EpoR-immunopositive particle count (68%, $P = 0.048$) (Fig 6) with comparable particle sizes (16%,

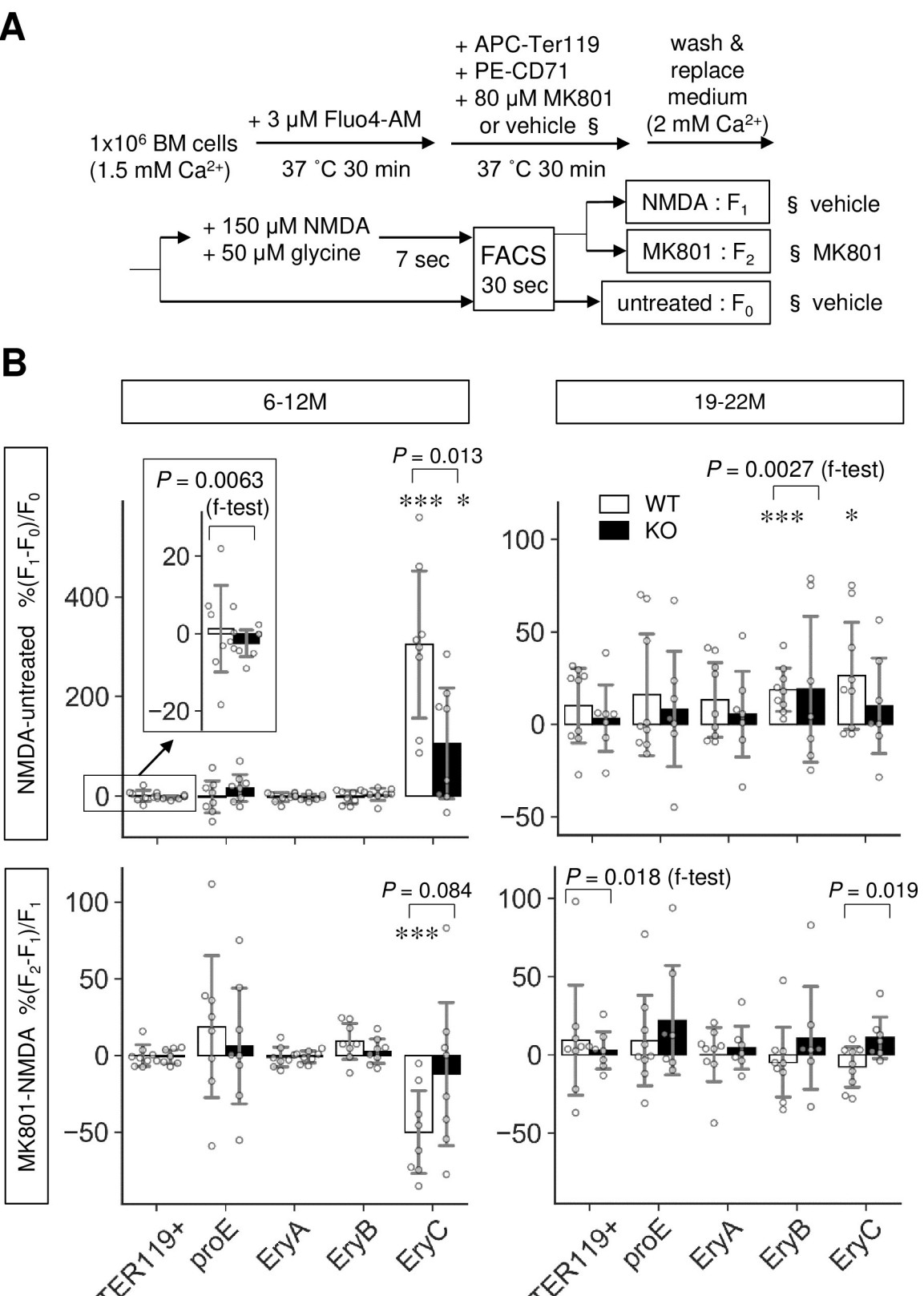

**Fig 5. Altered NMDA receptor-mediated Ca$^{2+}$ influx in Lrfn2 KO mice.** Bone marrow cells from 6–12 M-old (top; WT, $n$ = 8; KO, $n$ = 8) or 19–22 M-old (bottom; WT, $n$ = 9; KO, $n$ = 7) WT and Lrfn2 KO mice were subjected to Fluo4-based intracellular Ca$^{2+}$

measurement. (A) Outline of the assay. (B) %ΔF/F values between no treatment and NMDA/glycine treatment (*NMDA-untreated*) groups or those between NMDA/glycine treatment and NMDA/glycine treatment groups with MK-801 pretreatment (*MK801-NMDA*) are shown for total erythroblasts (TER119+), pro-erythroblasts (proE), early erythroblasts (EryA), mid erythroblasts (EryB), and late erythroblasts (EryC). *Open bar*, WT; *closed bar*, KO; *error bar*, SD. Each value from a mouse is indicated by *circles*. P values were obtained by two tailed *t*-tests between WT and KO groups. P values *(f-test)* were obtained by f-test between WT and KO. *, P < 0.05; ***, P < 0.001 in one tailed *t*-tests for null hypothesis denote mean values equal to zero.

P = 0.44) (Fig 6) and intensities (0.3%, P = 0.12). We also performed qPCR analysis using FACS fractionated proE, EryB, and EryC (S1 Fig). However, significant differences between WT and KO cells were not found in genes that critically regulate erythrocyte counts (*EpoR* and *Jak2*), encode NMDA receptor subunits (*Grin1* and *Grin2c*), or change in Lrfn2 KO EryC (*Tfrc*). *Jak2* mediates EpoR signaling and is a causal gene of polycythemia vera [19]. *Tfrc* encodes transferrin receptor (CD71), while *Grin1* and *Grin2c* encode the NMDA receptor subunits NMDAR1 and NR2C, respectively. Grin2c expression tended to be higher in late erythroblasts than in mid erythroblasts (S2 Fig). Histological analysis of BM and spleen did not reveal clear differences between WT and KO cells (S3 Fig).

## Discussion

This study revealed the importance of Lrfn2 in the regulation of erythropoiesis. Hemogram and BM cell analyses of Lrfn2 KO mice revealed multilineage abnormalities. In erythroid cells, normocytic erythrocythemia, decreased CFU-E, increased early erythroblast populations, and impaired late erythroblast differentiation were observed. In addition, we observed dysregulation of NMDA receptors involved in erythropoiesis.

Previous studies have elucidated the role of NMDA receptors in erythropoiesis. NMDA receptors have been detected in rat erythrocytes [20] and human erythroid precursor cells [21]. Moreover, NMDA receptors play a dual role in erythropoiesis, supporting survival of mid (EryB) erythroblasts and contributing to the $Ca^{2+}$ homeostasis in late (EryC) erythroblasts and circulating erythrocytes [15]. In a recent study using Meg-01 cells (models of human megakaryocytic and erythroid progenitors (MEP)), NMDAR1 KO (Meg-01-*GRIN1*$^{-/-}$) cells underwent atypical differentiation toward erythropoiesis, which is associated with increased basal ER stress and cell death [10]. Thus, NMDAR1 function may be involved in erythropoiesis at multiple stages, from the lineage determination to the late differentiation stage. In addition, calcium signaling, such as EPO signaling [22–24], PIEZO1 signaling [25], and erythroid enucleation [26], plays essential roles during erythroid differentiation.

The roles of NMDA receptors in erythropoiesis and binding of Lrfn2 to NMDAR1 [1] led us to hypothesize that the erythropoietic abnormalities in Lrfn2 KO were caused by alterations in NMDA receptor function or calcium signaling (Fig 7). First, the decreased CFU-E may reflect the impaired role of NMDAR1 in MEP [10]. This hypothesis is supported by increased BFU-E in Lrfn2-overexpressing BM cells [8]. Together with the BFU-E decrease tendency in Lrfn2 KO, CFU-E decrease might reflect the impaired differentiation from MEP to BFU-E. Otherwise, CFU-E decrease could be associated with intracellular calcium signaling after stimulation by interleukin-3 and granulocyte-macrophage colony-stimulating factor [27]. Second, increased early erythroblasts (EryA) may reflect impaired differentiation into EryB and/or EryC. Third, increased CD71 level in EryC may reflect impaired differentiation into EryC, because the expression of CD71 on the cell surface decreases during the differentiation of EryB to EryC in murine erythropoiesis [28]. NMDA receptor plays a dual role in the survival of EryB and maintenance of calcium homeostasis during the differentiation of EryC into circulating erythrocytes [15]. Collectively, erythropoietic abnormalities in Lrfn2 KO can be explained

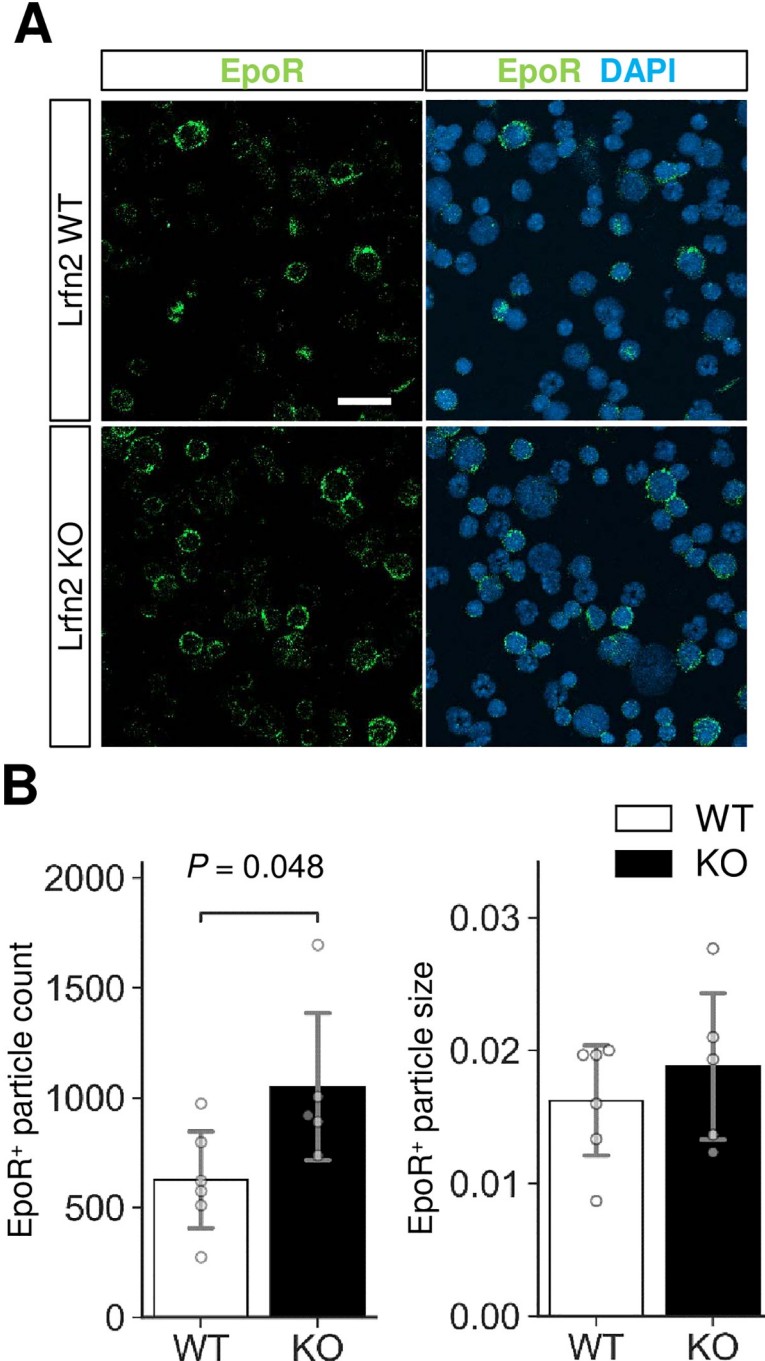

**Fig 6. Immunostaining of EPO receptor (EpoR).** BM cells were immunostained with anti-EpoR antibody (green). Nuclei were stained with DAPI (blue). (A) Representative images. Scale bar, 20 μm. (B) Quantification of the EpoR-immunopositive signals (WT, $n$ = 6 mice; KO, $n$ = 5 mice). *Open bar*, WT; *closed bar*, KO; *error bar*, SD. Each value from a mouse is indicated by *circles*. $P$ values were obtained by two tailed $t$-tests between WT and KO.

by assuming that NMDA receptor functions are altered in both the early phase (MEP stage), and the late phase (EryB, and EryC stages).

At the individual level, there was no difference between the numbers of cells in femur and erythroid (TER119[+]) cells in BM. This can explain the counteracting effects during MEP

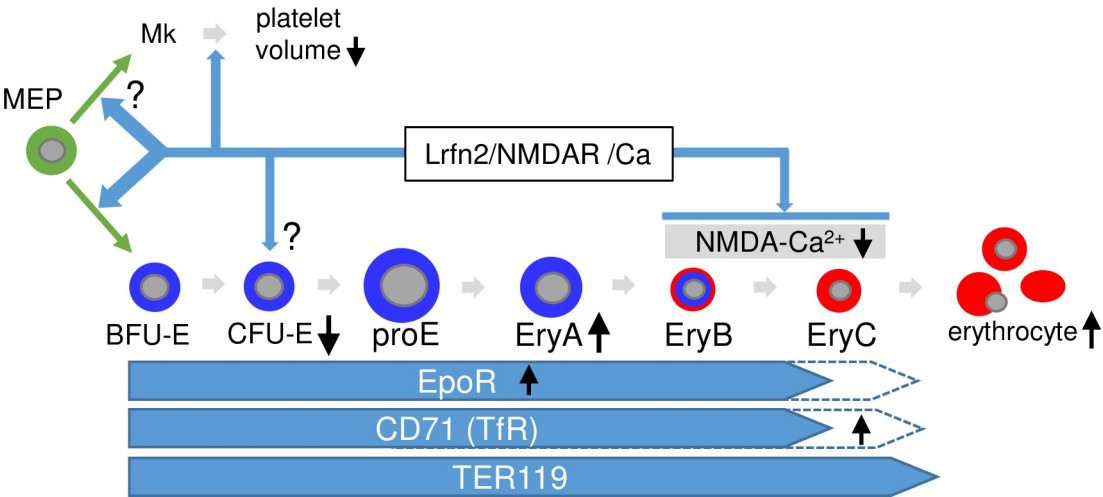

**Fig 7. Hypothetical model to explain the role of Lrfn2 in hematopoiesis.** Possible roles of Lrfn2 (top), differentiation stages of erythrocyte lineage (middle), and molecular marker expression profiles (bottom) are indicated. *Black vertical arrows* indicate the major abnormalities in Lrfn2 KO mice described in this paper. *MEP*, megakaryocyte-erythroid progenitor cell; *Mk*, megakaryocytes.

(decreased counts of CFU-E and BFU-E cells) and EryB and EryC stages (apparently increased counts of erythroid cells in BM due to delayed differentiation). However, we did not obtain direct evidence to explain the increased circulating erythrocytes number in Lrfn2 KO mice in this study. Nevertheless, increased EpoR expression in BM cells may be associated with erythrocytosis because EpoR is required to suppress apoptosis in later stages of erythroblast maturation [29]. Moreover, NMDA receptors in mature erythrocytes may affect the life span of erythrocytes. In mature human and rat erythrocytes, NMDA receptors impact cellular nitric oxide synthase activity [9,20]. It is possible that altered NMDA receptor function affects the life span of erythrocytes.

The decreased platelet volume in Lrfn2 KO mice indicates the involvement of Lrfn2 in thrombopoiesis, in addition to its role in erythropoiesis. NMDAR1 is expressed in human BM megakaryocytes [30], and MK-801 inhibits not only proplatelet formation in megakaryocytes but also megakaryocytic differentiation from hematopoietic stem cells [31]. In mice, thrombocytopenia has been described as a phenotype of NMDAR1 KO heterozygotes ($Grin1^{-/+}$) (Mouse Genome Informatics and the Wellcome Trust Sanger Institute Mouse Genetics Project, http://www.informatics.jax.org/allele/genoview/MGI:5781633). Therefore, the decreased platelet volume may also be explained by the altered function of NMDA receptor.

Although the above discussion focused on NMDA receptors to elucidate the hematopoietic role of Lrfn2, we cannot exclude other possibilities at this point. Considering the low abundance of both Lrfn2 and NMDA receptors in hematopoietic cells in comparison to those of brains, it would be necessary to further address the hematopoietic role of NMDA receptors using genetically modified mice for better understanding. To our knowledge, very few studies have addressed the role of the so-called synapse adhesion molecules in hematopoiesis. It would be interesting to investigate the roles of synaptic molecular complexes in hematopoiesis and physiological regulation of blood cells.

## Supporting information

**S1 Fig. Cell number and size of Lrfn2 KO bone marrow.** (left) BM cell number from a femur. (right) mean cell diameter. WT, n = 14; KO, n = 13 mice at 6–12 M-old male. Live BM

cells were analyzed after trypan blue staining using automated cell counter (TC20, Bio-Rad).
(PDF)

**S2 Fig. Quantitative PCR analysis of mRNA levels.** Total RNAs from FACS-fractionated bone marrow cells were subjected to analysis. Cells derived from 5 M-old male mice. Values were normalized to those of 18S ribosomal RNA. WT, $n$ = 3 mice; KO, $n$ = 3 mice. *Open bar*, WT; *closed bar*, KO; *error bar*, SD. Each value from a mouse is indicated by *circles*. *P* values were obtained by two tailed *t*-tests. BM cells were fractionated by BD FACSAria sorter (BD Biosciences). RNA was isolated from the BM cells using TRIzol Reagent (Thermo Fisher). cDNA was synthesized using SuperScript II Reverse Transcriptase (Thermo Fisher). Realtime RT-PCR analysis was carried out using Power SYBR Green PCR Master Mix (Thermo Fisher), QuantStudio 12K Flex Real-Time PCR System (Thermo Fisher).
(PDF)

**S3 Fig. Hematoxylin & Eosin staining of bone marrow and spleen sections.** Sections derived from 5–6 M-old male mice. Scale bar, 50 μm. The tissues were fixed by immersion in 4% PFA, 0.1 M sodium phosphate buffer, pH 7.4, overnight at 4 ˚C, dehydrated in a graded series of ethanol, cleared in xylene, and embedded in paraffin blocks for light microscopy. Six-micrometer-thick sections of tissues were cut with a microtome, mounted on slides, and stained with hematoxylin and eosin. Images were taken by BZ-X700 microscope (Keyence).
(PDF)

**S1 Table. Hemogram of Lrfn2 KO mice.** Complete blood count of male Lrfn2 WT and KO mice aged 3 M (2.0–4.3 M-old; WT, $n$ = 6; KO, $n$ = 7), 6 M (5.8–7.2 M-old; WT, $n$ = 5; KO, $n$ = 6), and 12 M (10.3–13.1 M-old; WT, $n$ = 16; KO, $n$ = 14). Results are indicated as the mean ± SD. Percentages indicate changes in KO mice compared to the mean in WT mice (WT mean = 100%). *Italicized* values indicate the P values obtained by two-tiered unpaired *t*-test. **Bold** letters indicate the percentages and P values with significant differences between WT and KO mice. WBC, leukocyte count; Lymph, lymphocyte count; Mono, monocyte count; Gran, granulocyte count; Lymph%, percentage of lymphocytes in total leukocytes; Mono%, percentage of monocytes in total leukocytes; Gran%, percentage of granulocytes in total leukocytes; RBC, erythrocyte count; HGB, hemoglobin concentration; HCT, hematocrit; MCV, mean corpuscular volume; MCH, mean corpuscular hemoglobin; MCHC, mean corpuscular hemoglobin concentration; RDW; red cell (erythrocyte) distribution width; PLT, platelet count; MPV, mean platelet volume; PDW, platelet distribution width; PCT, plateletcrit.
(PDF)

**S2 Table. Flow cytometry analysis of Lrfn2 KO BM cells.** BM cells from 18–22 M-old Lrfn2 WT ($n$ = 8) and KO ($n$ = 7) mice were subjected to CD71/TER119 flow cytometry. Results are indicated as the mean ± SD. Percentages indicate changes in KO mice, compared to the mean in WT mice (WT mean = 100%). *Italicized* values indicate the *P* value obtained by two-tiered unpaired *t*-test. **Bold** letters indicate the percentages and *P* values with significant differences between WT and KO. Cell count, numbers of cells out of 10,000 living cells (7AAD-negative); CD71, mean immunofluorescence intensity (arbitrary unit) given by anti-CD71-PE; TER119, mean immunofluorescence intensity (arbitrary unit) given by anti-Ter119-FITC; FSC, forward scatter area; SSC, side scatter area.
(PDF)

## Acknowledgments

We thank Dr. Kazuo Yamamoto (Biomedical Research Support Center, Nagasaki University) for his helpful advice.

## Author Contributions

**Investigation:** Ryuta Maekawa, Hideki Muto, Minoru Hatayama, Jun Aruga.

**Writing – original draft:** Jun Aruga.

**Writing – review & editing:** Jun Aruga.

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
