## [Decision Letter · Decision Letter 0]

16 Nov 2020

PONE-D-20-32238

Dysregulation of haematopoiesis and altered erythroblastic NMDA receptor-mediated calcium influx in Lrfn2-deficient mice

PLOS ONE

Dear Dr. Jun Aruga,

Thank you for submitting your manuscript to PLOS ONE. After careful consideration, we feel that it has merit but does not fully meet PLOS ONE’s publication criteria as it currently stands. Therefore, we invite you to submit a revised version of the manuscript that addresses the points raised during the review process.

Pleasr revise your paper taking into account the remarks of 3 reviewers and Academic Editor. The most important is to pay attention to progenitor's experiments data interpretation.

If you decided to perform the new experiments with in vivo repopulating assays, please submit your revised manuscript by August 2021. If you  decide to complete your revisions without adding new experiments, you have two months from the day of receiving the decision. Please informe the Editorial Board on your intentions by contacting the journal office at plosone@plos.org. Please include the following items when submitting your revised manuscript:

We look forward to receiving your revised manuscript.

Kind regards,

Zoran Ivanovic, MD, PhD, HDR

Academic Editor

PLOS ONE

Journal Requirements:

Additional Editor Comments (if provided):

The results are convincing for CFU-E but for other committed progenitors (CFU-GM, CFU-GEMM) the amplitude of differences is rather low. Thus, the primary effect concerns late erythropoietic progenitors and erythropoiesis.

Even if CFU-GM numbers were decreased it does not mean automatically « impaired development of granulocyte progenitors » but could come from the lower production of these progenitors from more primitive stem/progenitor cells, enhanced differentiation and consequent exhaustion of CFU-GM compartment, or both.

So, if the claim concerning « hematopoiesis « is maintained it is necessary to explore the stem cell compartment (for example by MRA assay or some other repopulation test revealing the stem cell activity.).

It would be more appropriate show the progenitor’s data as the number of progenitors per femur. In any case it is necessary to give the number of cells plated in the Petri dishes or at least which volume fraction of femoral bone marrow suspension was plated.

Reviewers' comments:

Reviewer's Responses to Questions

**Comments to the Author**

1. Is the manuscript technically sound, and do the data support the conclusions?

Reviewer #1: Partly

Reviewer #2: Yes

Reviewer #3: Partly

2. Has the statistical analysis been performed appropriately and rigorously? 

Reviewer #1: N/A

Reviewer #2: I Don't Know

Reviewer #3: I Don't Know

3. Have the authors made all data underlying the findings in their manuscript fully available?

Reviewer #1: Yes

Reviewer #2: Yes

Reviewer #3: Yes

4. Is the manuscript presented in an intelligible fashion and written in standard English?

Reviewer #1: Yes

Reviewer #2: No

Reviewer #3: No

5. Review Comments to the Author

Reviewer #1: The Manuscript “Dysregulation of haematopoiesis and altered erythroblastic NMDA receptor-mediated calcium influx in Lrfn2-deficient mice” provides interesting insights potentially significant for the research topic. In general, the manuscript is well-written, but there are several issues:

Figure 3. Why Authors did not analyze splenic colonies (both erythroid and myeloid) when observed increase in spleen weight?

Figure 4.A Could Authors provide some plots where the differences in the erythroblast populations are more convincing? Also, it would be more informative to have the percentages of cells instead of MFIs here. In that context, please be careful when say “increased early erythroblast populations” (Discussion, line3-4).

Discussion: Please discuss on basis on obtained results, but in more critical manner (Particularly page 19). E.g. “Decreased leukocyte number and GM-CFU suggest impaired development of granulocyte progenitors. Decreased platelet volume suggests impaired platelet production.“ – please comment and discuss the results more in detail. Also, In general, the significance of calcium signaling for erythroid cells remained unexplained completely.

Figure 7. Shematic illustration is very useful, but on basis on the results presented here, authors can only speculated regarding significance of Lrnf2 for erythroid lineage cells, while for the others, it would be necessary to involve more phenotype and functional markers in the research.

Minor comments:

Please uniform the labeling of colony types (e.g. CFU-GM throughout the paper) and indicate at which time point CFU-Es were scored. Also, when interpret BFU-E/CFU-E progenitors, be more precise and uniform in term of their primitiveness (Page 11, 19)

Please specify whether RBC lysis was performed for BM cell evaluation.

Reviewer #2: The paper titled “Dysregulation of hematopoiesis and altered erythroblastic NMDA receptor-mediated calcium influx in Lrfn2-deficient mice” has an attempt to elucidate the role of two receptors in erythropoiesis. The subject is interesting and in the first part of the manuscript very promising, but discussion has failed to underline the most important points in this study. However, I think that it could rewritten for better explanation of results.

Except the discussion, I would like to drive the attention of the authors to the section Material and methods and to ask them to provide the total number of animals used, the number of animals used for each separate part of the experiment, to provide information about volumes of blood obtained per animal, info about possibility of pooling the samples, and finally to provide the algorithm of the experimental design. Also, it is highly recommended that authors provide the statistical method how they calculated the number of animals needed for the study.

Discussion

Unfortunately the discussion does not seem completely focused and is a bit confusing.

First sentence suggests that Lrfn2 is essential for hematopoiesis but I would say that it is better to use “important”

The second paragraph does not discuss the results of the authors and it seems to be to long and unnecessary for the discussion. It was already mentioned that NMDR has an important role in neuronal development. However, if the authors want to discuss the important of this molecules in neuronal function, it seems as a better solution to explain final functional dysregulation that arise as a consequence of Lrfn2 overexpression and absence.

Although the third paragraph discuss the effect of NMDR absence on thrombopoiesis it does not discuss the results of the authors, and at this point I would like to suggest that each paragraph should discuss one of the results that authors obtained.

Paragraph 4 – I suggest not to use reference to the figures in discussion.

Paragraph 4 and 5 seems to be the most important ones, so I suggest authors to focus to these paragraphs to drive out the most of their discussion. Calcium signaling and hypoxia are very important for regulation of erythropoietic response so it would be nice to address this question more deeply.

Reviewer #3: General comment

The work by Ryuta Maekawa et al. aims to evaluate the impact of Lrfn2 gene knockout on hematopoiesis in mouse model and to understand its mechanism of action notably through the receptor NMDA. The question is interesting, the study is relatively well designed, and the results are presented in a very honest way in the course of the manuscript. However, the manuscript lacks of consistency because of apparent low effect of Lrfn2 knockout and because of manuscript writing. Indeed, in general, the differences between WT and KO Lrfn2 are weak, the explanations are rapid and they lack of details, the link between experiments and between results are not clear enough, and the conclusions look too much extrapolated. Therefore, it is difficult here to understand the real input of the present work.

Please find below some comments/propositions that may potentially help.

Major comments

“Results” section

Results relative to Figure 1:

- What is the biological significance of the in mice? Do the KO and its biological consequences (increase of erythrocytes counts, hematocrit, hemoglobin and decrease of platelets volume and leukocyte count) lead to symptomatic “disease” or modification of efficiency regarding O2/CO2 transport and hemostasis in mice? In general, is there an impact of Lrfn2 KO on the well-being of mice?

- On 5th panel (leukocyte count), man can observe a 50% decrease of the number of leukocytes in WT mice according to the age? Is that decease expected? Could the authors explain that decrease and/or cited relevant bibliography dealing that point in order to dissipate potential questioning by the readers?

Results relative to Figure 3:

- Here, the conclusion (last sentence page 12) is mainly based on the divergence of rates weight between WT and KO mice. That conclusion is too rapid; it is difficult to understand how Lrfn2 could directly affect the late stage of erythropoiesis (in spite of the citation of reference [8]).

Results relative to Figure 4:

- Figure 4 shows that (i) the number of total erythroid cells (TER119+) is not different between WT and KO mice, (ii) only the number of EryA is slightly increased in KO mice, the CD71 is slightly overexpressed in EryC only, and the SSC is slightly decreased in ProE. Therefore, the authors conclude that KO Lrfn2 affects both early and late phase of erythropoiesis. That conclusion is very rapid and needs some precisions. Indeed, because for each parameter measured, the Lrfn2 KO affects erythroid cells at different stages of maturation, the authors have to explain better the result and conclude in appropriate way. For example, what means the decrease of CD71 receptor in EryC? What means the decrease of SSC in proE?

- The fact that the number of total erythroid cells remains unchanged between WT and KO is little in contrast with both the decrease of CFU-E (Figure 2) and the increase of erythrocyte count (Figure 1). That point has to be addressed in the conclusion in order to dissipate any confusions.

Results relative to Figure 5:

- In general, the organization of the results in the text has to be improved (e.g. better organized) in order to facilitate the comprehension; in the current state, the results are not easy to catch (which is even more a shame since it probably concerns the most relevant and interesting data of the manuscript).

Results relative to Figure 6:

- Concerning the following sentence “In immunostaining of BM cells we observe significant increase of EpoR-immunopositive particle count (16%, P=0.44) and intensity (0.3%, P=0.12)”, despite figure 6 is clear regarding the data presented, the authors have to indicate in the text in which condition they observe that increase.

- The qPCR analysis is relevant; however, the authors have to explicit better the rational of the analyzed genes (i.e. why Jak2, Tfrc, Grin1, etc… are chosen?). Even it is implicit, it has to be more precisely indicated in the text.

“Discussion” section

- The hypothesis of the role of hypoxia downstream the EpoR and its modulation by NMDA receptor needs to be better supported.

- In figure 7, the authors tried to gather the experimental data, which is welcome. Unfortunately, it is not sufficient to clarify the manuscript in general and to fill the lack in the text.

Minor comments

“Abstract” section (page 3)

The following sentence is confusing and has to be clarified or reformulated: “We also found that Lrfn2 KO late erythroblasts were defective in NMDA receptor-mediated calcium influx and its inhibition by an NMDA receptor antagonist MK801”.

“Results” section

Results relative to Figure 1:

- For easier reading of the manuscript, please indicate panel A, B, C, D and E on figure 1 and refer to this nomenclature in the text.

6. PLOS authors have the option to publish the peer review history of their article (what does this mean?). If published, this will include your full peer review and any attached files.

Reviewer #1: No

Reviewer #2: No

Reviewer #3: No

---

## [Author Response · Author response to Decision Letter 0]

1 Jan 2021

Major points of revision:

1. The title was changed into “Dysregulation of erythropoiesis and altered erythroblastic NMDA receptor-mediated calcium influx in Lrfn2-deficient mice”

2. We added supplementary Fig. S1 that show the cell yield from a femur and mean diameter of bone marrow cells.

3. We carried out the colony forming units assay using spleen cells. The results were added in Fig. 2.

4. We indicated body weight and (spleen weight)/(body weight) of Lrfn2 KO in Fig. 3.

5. Discussion was rewritten and Fig. 7 were reorganized and the text was corrected by a science text editor. 

Point-by-point responses to reviewers’ comments:

Editor:

The results are convincing for CFU-E but for other committed progenitors (CFU-GM, CFU-GEMM) the amplitude of differences is rather low. Thus, the primary effect concerns late erythropoietic progenitors and erythropoiesis.

Even if CFU-GM numbers were decreased it does not mean automatically « impaired development of granulocyte progenitors » but could come from the lower production of these progenitors from more primitive stem/progenitor cells, enhanced differentiation and consequent exhaustion of CFU-GM compartment, or both.

So, if the claim concerning « hematopoiesis « is maintained it is necessary to explore the stem cell compartment (for example by MRA assay or some other repopulation test revealing the stem cell activity.).

It would be more appropriate show the progenitor’s data as the number of progenitors per femur. In any case it is necessary to give the number of cells plated in the Petri dishes or at least which volume fraction of femoral bone marrow suspension was plated.

Thank you for the helpful comments.

According to the suggestion, we removed the word or phrase for meaning “haematopoiesis abnormality”, and removed the discussion concerning CFU-GM. We indicated the number of bone marrow cells from a femur in a new supplementary Fig. S1. The y-axis scale of CFU assay graph was changed to indicate the colony numbers from 1×104 cells, and the number of the plated cells in the CFU assays were indicated in Materials and methods. 

Reviewer #1: 

Figure 3. Why Authors did not analyze splenic colonies (both erythroid and myeloid) when observed increase in spleen weight?

According to the reviewer’s suggestion, we carried out CFU assays using spleen cells. The results indicating changes similar to those of bone marrow CFUs were placed at the bottom of Fig. 2. 

Figure 4. A Could Authors provide some plots where the differences in the erythroblast populations are more convincing? Also, it would be more informative to have the percentages of cells instead of MFIs here. In that context, please be careful when say “increased early erythroblast populations” (Discussion, line3-4).

We indicated the cell number in Fig. 4B that corresponds to Fig. 4A. Cell number percentages are indicated in the scatter plots. Although we tried anther style (histogram), we think current presentation style is the best one. 

Discussion: Please discuss on basis on obtained results, but in more critical manner (Particularly page 19). E. g. “Decreased leukocyte number and GM-CFU suggest impaired development of granulocyte progenitors. Decreased platelet volume suggests impaired platelet production.“ – please comment and discuss the results more in detail. Also, In general, the significance of calcium signaling for erythroid cells remained unexplained completely.

We revised the discussion entirely. The discussion concerning the granulopoiesis was removed from this paper because of its uncertainty (please see above editor comment and our response to it). We supplemented some roles of calcium signaling in erythropoiesis as follows: “In addition, calcium signalling is essential play essential roles during erythroid differentiation such as EPO signalling [22-24], PIEZO1 signalling [25], and erythroid enucleation [26].”

Figure 7. Schematic illustration is very useful, but on basis on the results presented here, authors can only speculate regarding significance of Lrfn2 for erythroid lineage cells, while for the others, it would be necessary to involve more phenotype and functional markers in the research.

We agree with this comment. We removed the granulopoiesis relevant part from Fig. 7 and corrected it to be logically consistent with the description in Discussion. 

Minor comments:

Please uniform the labeling of colony types (e.g. CFU-GM throughout the paper) and indicate at which time point CFU-Es were scored. Also, when interpret BFU-E/CFU-E progenitors, be more precise and uniform in term of their primitiveness (Page 11, 19)

We corrected “GM-CFU” into “CFU-GM”. CFU-Es were scored at 3 days in vitro, which was indicated in Materials and methods. We described the BFU-E/CFU-E interpretation as follows: “A recent study involving transcriptome analysis showed that BFU-E and CFU-E represent CD45+GPA-IL-3R-CD34+CD36-CD71low cells and CD45+GPA-IL-3R-CD34-CD36+CD71high cells, respectively, and suggested that erythroid differentiation progresses in the following order: CD34+ hematopoiesis progenitor cells → BFU-E cells → CFU-E cells → pro-erythroblasts [17]. Thus, the colony forming unit assay demonstrated that Lrfn2 KO mice exhibited either reduced counts of BFU-E cells or accelerated differentiation to pro-erythroblast, or both, irrespective of the increased count of mature erythrocytes. In any case, it was thought that early phase of erythropoiesis was impaired in Lrfn2 KO mice.”

Please specify whether RBC lysis was performed for BM cell evaluation.

We did not perform RBC lysis. This point was added in Materials and methods.

Reviewer #2: 

The paper titled “Dysregulation of hematopoiesis and altered erythroblastic NMDA receptor-mediated calcium influx in Lrfn2-deficient mice” has an attempt to elucidate the role of two receptors in erythropoiesis. The subject is interesting and in the first part of the manuscript very promising, but discussion has failed to underline the most important points in this study. However, I think that it could rewritten for better explanation of results.

We really appreciate below comments to improve the quality of this paper. Discussion and Materials and methods were revised according to the comments.

Except the discussion, I would like to drive the attention of the authors to the section Material and methods and to ask them to provide the total number of animals used, the number of animals used for each separate part of the experiment, to provide information about volumes of blood obtained per animal, info about possibility of pooling the samples, and finally to provide the algorithm of the experimental design. Also, it is highly recommended that authors provide the statistical method how they calculated the number of animals needed for the study.

The total number of animals and the number used in each experiment were indicated in Materials and methods section and each Figure legend. We also added the basis of sample number and the detailed description of experimental flow as follows: “A total of 67 WT and 67 KO mice were used in this study. Each sample was derived from a single mouse without pooling. Common mice were used partly between the peripheral blood test and the other tests. Common mice were used between spleen weight measurement and (calcium measurement test or colony assay). Sample sizes for each experiment were determined such that the power and significance in two sided-test were 80% and 5%, respectively [11]. However, the number of samples from animals were minimized empirically.” Blood volume used for CBC (100-200 μL) was indicated in Materials and methods.

Discussion

Unfortunately, the discussion does not seem completely focused and is a bit confusing.

First sentence suggests that Lrfn2 is essential for hematopoiesis but I would say that it is better to use “important”

We corrected it as suggested.

The second paragraph does not discuss the results of the authors and it seems to be too long and unnecessary for the discussion. It was already mentioned that NMDR has an important role in neuronal development. However, if the authors want to discuss the importance of this molecule in neuronal function, it seems as a better solution to explain final functional dysregulation that arise as a consequence of Lrfn2 overexpression and absence.

We removed paragraph 2.

Although the third paragraph discuss the effect of NMDR absence on thrombopoiesis it does not discuss the results of the authors, and at this point I would like to suggest that each paragraph should discuss one of the results that authors obtained.

The paragraph was divided into two parts so that logical flow about erythropoiesis becomes clear. Following paragraphs were also reorganized according to the suggestion.

Paragraph 4 – I suggest not to use reference to the figures in discussion.

We removed the reference to Fig. 5, and remained that to Fig. 7. We think the reference to Fig. 7 is necessary in this case.

Paragraph 4 and 5 seems to be the most important ones, so I suggest authors to focus to these paragraphs to drive out the most of their discussion. Calcium signaling and hypoxia are very important for regulation of erythropoietic response so it would be nice to address this question more deeply.

Accordingly, we supplemented new references and extended the discussion on calcium signaling in erythropoiesis. 

Reviewer #3: 

The work by Ryuta Maekawa et al. aims to evaluate the impact of Lrfn2 gene knockout on hematopoiesis in mouse model and to understand its mechanism of action notably through the receptor NMDA. The question is interesting, the study is relatively well designed, and the results are presented in a very honest way in the course of the manuscript. However, the manuscript lacks of consistency because of apparent low effect of Lrfn2 knockout and because of manuscript writing. Indeed, in general, the differences between WT and KO Lrfn2 are weak, the explanations are rapid and they lack of details, the link between experiments and between results are not clear enough, and the conclusions look too much extrapolated. Therefore, it is difficult here to understand the real input of the present work.

Please find below some comments/propositions that may potentially help.

Major comments

“Results” section

Results relative to Figure 1:

- What is the biological significance of the in mice? Do the KO and its biological consequences (increase of erythrocytes counts, hematocrit, hemoglobin and decrease of platelets volume and leukocyte count) lead to symptomatic “disease” or modification of efficiency regarding O2/CO2 transport and hemostasis in mice? In general, is there an impact of Lrfn2 KO on the well-being of mice?

We newly added the body weight of 10-12 M-old and 19-27 M-old mice in Fig. 3B. The body weight at 10-12 M-old in comparison to littermate WT was -14% (P=0.00054), stronger change than that observed in 2M old mice (-6.0%, P = 0.022). We think this phenotype partly reflects the blood abnormality. We also normalized the spleen weight by body weight and found unexpectedly strong difference in the divergences between WT and KO. The result is shown in Fig. 3B and described as follows: “In both the 10-12 M-old and 19-27 M-old age groups, the average weights of spleens were comparable between WT and KO mice (Fig 3B). However, the weights of spleen from KO mice were less divergent than from WT mice in the 10-12 M-old age group (P = 9.0 ×10-5 in f-test) but more divergent in the 19–27 M-old age group (P = 0.041 in f-test) (Fig 3B), indicating hematological dysregulation that is distinct from polycythemia vera.”

- On 5th panel (leukocyte count), man can observe a 50% decrease of the number of leukocytes in WT mice according to the age? Is that decease expected? Could the authors explain that decrease and/or cited relevant bibliography dealing that point in order to dissipate potential questioning by the readers?

This is an interesting point. We examined the effects of aging on leukocyte numbers in bibliography. However, we did not see relevance papers. Although the basis of this large decrement is unclear, the comparison between WT and KO seemed meaningful. With respect to the discussion on the granulopoiesis phenotype, we totally removed from this paper because the phenotype is not clear as pointed out by the editor. 

Results relative to Figure 3:

- Here, the conclusion (last sentence page 12) is mainly based on the divergence of rates weight between WT and KO mice. That conclusion is too rapid; it is difficult to understand how Lrfn2 could directly affect the late stage of erythropoiesis (in spite of the citation of reference [8]).

Please see the above response to Figure 1. To clearly indicate logical flow, the last part was rewritten as follows: “.., indicating hematological dysregulation that is distinct from polycythemia vera. The increased serum EPO level, together with Lrfn2 overexpression [8], led us to hypothesize that Lrfn2 deficiency directly affects erythropoiesis in vivo.” 

Results relative to Figure 4:

- Figure 4 shows that (i) the number of total erythroid cells (TER119+) is not different between WT and KO mice, (ii) only the number of EryA is slightly increased in KO mice, the CD71 is slightly overexpressed in EryC only, and the SSC is slightly decreased in ProE. Therefore, the authors conclude that KO Lrfn2 affects both early and late phase of erythropoiesis. That conclusion is very rapid and needs some precisions. Indeed, because for each parameter measured, the Lrfn2 KO affects erythroid cells at different stages of maturation, the authors have to explain better the result and conclude in appropriate way. For example, what means the decrease of CD71 receptor in EryC? What means the decrease of SSC in proE?

Together with the comments from the other reviewer’s, we entirely rewrote the discussion regarding Figure 4. Detailed description in Result section was not possible because following results are needed for better interpretation of the results. Concerning the increased CD71, we described as follows: “Third, increased CD71 level in EryC may reflect impaired differentiation into EryC, because the expression of CD71 on the cell surface decreases during the differentiation of EryB to EryC in murine erythropoiesis [28].” The decrease of SSC in proE was not discussed because the change was mild and not significant.

- The fact that the number of total erythroid cells remains unchanged between WT and KO is little in contrast with both the decrease of CFU-E (Figure 2) and the increase of erythrocyte count (Figure 1). That point has to be addressed in the conclusion in order to dissipate any confusions.

We hypothesized that this is because the Lrfn2 and presumptive alteration of NMDA receptor-calcium signaling can influence at multiple stages (MEP, EryB, and EryC) of the erythroid differentiation. The absence of them at MEP decreases CFU-E and BFU-E whereas the absence of them at EryB and EryC stage prolong the time required for differentiation in BM. This point was described in Discussion as follows: “At the individual level, there was no difference between the numbers of cells in femur and erythroid (TER119+) cells in BM. This can explain the counteracting effects during MEP (decreased counts of CFU-E and BFU-E cells) and EryB and EryC stages (apparently increased counts of erythroid cells in BM due to delayed differentiation).” 

Results relative to Figure 5:

- In general, the organization of the results in the text has to be improved (e.g. better organized) in order to facilitate the comprehension; in the current state, the results are not easy to catch (which is even more a shame since it probably concerns the most relevant and interesting data of the manuscript).

To improve the readability, we divided the long paragraph into small ones and text was edited again by asking a scientific text editing service.

Results relative to Figure 6:

- Concerning the following sentence “In immunostaining of BM cells we observe significant increase of EpoR-immunopositive particle count (16%, P=0.44) and intensity (0.3%, P=0.12)”, despite figure 6 is clear regarding the data presented, the authors have to indicate in the text in which condition they observe that increase.

This may have occurred in the file conversion system. The original text was as follows:“In immunostaining of BM cells, we observed significant increase of EpoR-immunopositive particle count (68%, P = 0.048) (Fig 6) with comparable particle size (16%, P = 0.44) (Fig 6) and intensity (0.3%, P = 0.12).”We will re-check this point.

- The qPCR analysis is relevant; however, the authors have to explicit better the rational of the analyzed genes (i.e. why Jak2, Tfrc, Grin1, etc… are chosen?). Even it is implicit, it has to be more precisely indicated in the text.

We added following brief explanation before the explanation with references “However, significant differences between WT and KO cells were not found in genes that critically regulate erythrocyte counts (EpoR and Jak2), encode NMDA receptor subunits (Grin1 and Grin2c), or change in Lrfn2 KO EryC (Tfrc).”

“Discussion” section

- The hypothesis of the role of hypoxia downstream the EpoR and its modulation by NMDA receptor needs to be better supported.

This point was removed to reorganize the discussion to be focused on the interpretation of current results. 

- In figure 7, the authors tried to gather the experimental data, which is welcome. Unfortunately, it is not sufficient to clarify the manuscript in general and to fill the lack in the text.

We revised the Figure 7 to exclude the reference to “granulopoiesis associated phenotype” and to be more fitted in introducing our hypothesis. 

Minor comments

“Abstract” section (page 3)

The following sentence is confusing and has to be clarified or reformulated: “We also found that Lrfn2 KO late erythroblasts were defective in NMDA receptor-mediated calcium influx and its inhibition by an NMDA receptor antagonist MK801”.

We rewrote as follows: “Further, we found that late erythroblasts in Lrfn2 KO exhibited defective NMDA receptor-mediated calcium influx, which was inhibited by the NMDA receptor antagonist MK801.”

“Results” section

Results relative to Figure 1:

- For easier reading of the manuscript, please indicate panel A, B, C, D and E on figure 1 and refer to this nomenclature in the text.

We revised it accordingly.

---

## [Editor Report · Decision Letter 1]

5 Jan 2021

Dysregulation of erythropoiesis and altered erythroblastic NMDA receptor-mediated calcium influx in Lrfn2-deficient mice

PONE-D-20-32238R1

Dear Dr. Aruga,

We’re pleased to inform you that your manuscript has been judged scientifically suitable for publication and will be formally accepted for publication once it meets all outstanding technical requirements.

Kind regards,

Zoran Ivanovic, 

Academic Editor

PLOS ONE
---

## [Editor Report · Acceptance letter]

12 Jan 2021

PONE-D-20-32238R1 

Dysregulation of erythropoiesis and altered erythroblastic NMDA receptor-mediated calcium influxin Lrfn2-deficient mice 

Dear Dr. Aruga:

I'm pleased to inform you that your manuscript has been deemed suitable for publication in PLOS ONE. Congratulations! Your manuscript is now with our production department. 

Kind regards, 

on behalf of

Dr. Zoran Ivanovic 

Academic Editor

PLOS ONE